# Usefulness of a Colonoscopy Cap with an External Grid for the Measurement of Small-Sized Colorectal Polyps: A Prospective Randomized Trial

**DOI:** 10.3390/jcm10112365

**Published:** 2021-05-27

**Authors:** Seul-Ki Han, Hyunil Kim, Jin-woo Kim, Hyun-Soo Kim, Su-Young Kim, Hong-Jun Park

**Affiliations:** 1Department of Internal Medicine, Wonju College of Medicine, Yonsei University, Wonju 24626, Korea; lolhsk@naver.com (S.-K.H.); kimhyunil@empas.com (H.K.); bluebloood@naver.com (J.-w.K.); hyskim@yonsei.ac.kr (H.-S.K.); breeze1212@yonsei.ac.kr (S.-Y.K.); 2Center of Evidence Based Medicine, Institute of Convergence Science, Yonsei University, Wonju 24626, Korea

**Keywords:** colonic polyp, resection, colorectal cancer

## Abstract

Accurate measurement of polyp size during colonoscopy is crucial. The usefulness of cap-assisted colonoscopy and external grid application on monitor (gCAP) was evaluated for polyp size measurement in this 3-year, single-center, single-blind, randomized trial. Using the endoscopic forceps width as reference, the discrepancy percent (DP), error rate (ER), and measurement time were compared between gCAP and visual estimation (VE) after randomization. ER was calculated within a 20% and 33% limit. From the 111 patients, 280 polyps were measured. The mean polyp sizes were 4.0 ± 1.7 mm and 4.2 ± 1.8 mm with gCAP and VE, respectively (*p* = 0.368). Compared with that by the forceps method, DP was significantly lower in the gCAP group than in the VE group. Moreover, ER was significantly lower in the gCAP group within its preset limit. The measurement time was 4 s longer in the gCAP group than in the VE group (8.2 ± 4.8 s vs. 4.2 ± 1.5 s; *p* < 0.001). However, the forceps method lasted 28 s longer than the others. On subgroup analysis by size, gCAP was more accurate for polyp size ≥ 5 mm. The gCAP method was more accurate for polyp size measurement than VE, especially for polyps ≥ 5 mm, and was more convenient than the forceps method.

## 1. Introduction

Accurate measurement of the polyp size is crucial for screening and surveillance in colorectal cancer [1,2]. Colonic polyp size is a risk factor for colorectal cancer, with large polyps associated with a greater risk than smaller ones. Therefore, adenomas of more than 1 cm require shorter follow-up intervals than the smaller adenomas. Hence, incorrect measurement of a polyp size can affect its follow-up interval [3].

Additionally, the recommended removal techniques for small and diminutive polyps are different from those for large polyps. For example, cold forceps polypectomy is recommended for polyps of 1–3 mm, while cold snare polypectomy is recommended for polyps of 4–7 mm and sessile polyps of 6–9 mm [4]. 

Although pathologic size measurement has been the “gold standard”, polyp size is usually measured by subjective visual estimation (VE). However, VE of polyp size is inaccurate, and its discrepancy with the pathologic size is substantial [5,6,7,8]. Furthermore, the polypectomy technique should be dependent on colonoscopy. Therefore, accurate measurement of polyp size during colonoscopy is pivotal. To increase clinical accuracy, polyps are measured using various tools, such as an endoscopic ruler, forceps, a graduated needle, and a calibrated colonic cap, but these tools can be time-consuming and expensive [9,10,11,12].

Cap-assisted colonoscopy has higher polyp detection and cecal incubation rates than standard colonoscopy [13,14]. In this study, we used a colonoscopic cap and a transparent grid with 1-mm-spaced grid lines. The grid paper was placed on the monitor for more precise polyp size measurement (grid cap; gCAP). This study was conducted to evaluate and compare the accuracy and clinical convenience (feasibility) of polyp size measurement between the gCAP and VE methods.

## 2. Materials and Methods

This was a single-blind (participants), randomized, single-center trial with a parallel design. Patients were assigned to two groups in a 1:1 ratio, and the study involved four tertiary hospitals. This study was approved by the Institutional Review Board of Wonju Severance Christian Hospital (IRB number: CR312026) and was registered on clinical research information service (CRIS number: KCT0000754).

### 2.1. Subjects and Randomization 

Among patients aged 19–75 years who had visited Wonju Severance Hospital from July 2012 to June 2015 and had planned to undergo colonoscopy, patients who agreed to participate in the study and provided informed consent were considered eligible for this study. Patients with polyps of size 1–11 mm, detected during colonoscopy, were enrolled.

Patients were randomly assigned to either the gCAP or VE groups according to a computer-generated random number table. Central allocation by telephone was used to conceal the allocation.

### 2.2. Procedures and Intervention

Five endoscopists from one tertiary hospital participated in the study. All patients underwent colonoscopy under midazolam or propofol sedation, with or without opioid analgesics. All the colonoscopic procedures were performed using a standard scope (Q260, Olympus, Japan).

The inner diameter of the colonoscopic cap is approximately 11 mm. The grid was drawn on a transparent vinyl paper with 1-mm intervals (Figure 1A), and the paper was fixed at the inner circle of the colonoscopic cap (Figure 1B,C). At polyp detection, an assistant would place the grid on the monitor.

In the gCAP group, a colonoscope with a transparent hood cap attached to its distal tip was used. On the detection of a polyp, the endoscopist would attach the cap and measure the polyp size by counting on the external grid (Figure 1D,E). 

In both the groups, polyp size was additionally measured using forceps (open forceps width, 7 mm) to compare the gCAP and VE methods (Figure 1F).

### 2.3. Outcomes and Definitions

The primary outcome was accurate polyp size measurement. The discrepancies between gCAP (gCAP–forceps) and VE (VE–forceps) were compared. Additionally, the discrepancy percent (DP) was compared between the two groups. DP was calculated with the formula shown in Figure 2.

In addition, the error rate (ER) was compared between the groups. Error was defined as excessive DP more than limitation of DP (20% or 33%). A 20% discrepancy indicated a 1-mm error in a 5-mm-sized polyp, and a 33% discrepancy indicated a 1-mm error in a 3-mm-sized polyp. Overcall and undercall were defined as the overestimation and underestimation, respectively, of polyp size by gCAP or VE vs. that by forceps.

Although forceps measurement was used as a reference method, this method is not the gold standard method for exact polyp measurement [15,16]. Thus, the forceps measurements were recalculated using a graphic program. 

The secondary outcome was the measurement time required for the gCAP method compared with that for VE and forceps. Before each measurement, the endoscopist would freeze the endoscopic monitor and unfreeze the screen after the measurement. This was repeated for the VE group. The time from the freezing to the unfreezing was determined using a stopwatch, and this time was defined as the polyp measurement time. With forceps, measurement time was defined as the time from polyp detection to polyp withdrawal using forceps.

### 2.4. Statistical Analysis

This study was designed as a superiority test, and the independent variable was the method of polyp size measurement. The alpha, beta, and power were defined as 0.05, 0.20, and 0.80, respectively. The test was two-tailed, and the allocation ratio, N2/N1, was 1. Based on a previous study [17], the discrepancy in the gCAP and VE group was assumed to be 5% and 15%, respectively. The effect size(s) in both groups was 0.1. Therefore, the sample size was 140 polyps per group. The rate of loss to follow-up was 0%; therefore, a total of 280 polyps were evaluated. An interim analysis was not performed.

For primary and secondary outcomes, continuous data are expressed as medians with ranges or SDs, while categorical parameters are expressed as frequencies and proportions. For continuous data, parametric variables were compared using the independent *t*-test. Categorical parameters were compared using the χ^2^ test or Fisher’s exact test. All *p*-values were two-sided and were significant if *p* < 0.05. All statistical analyses were performed using SPSS software (version 25.0; SPSS Inc, Armonk, NY, USA). Subgroup analysis according to polyp size was performed. No interim analysis was performed.

## 3. Results

Of the 254 patients, 111 patients who had colorectal polyps 1–11 mm were enrolled. A total of 280 polyps were measured in the 111 patients (140 polyps each were measured by the gCAP and VE methods) (Figure 3).

The mean age of the enrolled patients was 64 years (gCAP vs. VE: 63.7 ± 85 years vs. 63.8 ± 10.9 years, *p* = 0.971), and the ratio of men to women was higher in the gCAP group than in the VE group (71.4% vs. 50.9%, *p* = 0.027). No differences in the ratio of expert endoscopists, history of abdominal surgery, bowel preparation, cecal intubation time, polyp withdrawal time, and number of detected polyps between the groups were noted.

Sedation was performed for more patients in the VE group than in the gCAP group (66.1% vs. 92.7%, *p* = 0.001). The patient characteristics are summarized in Table 1.

### 3.1. Mean Polyp Size, DP, and ER

The mean estimated polyp size was 4.0 ± 1.7 mm and 4.2 ± 1.7 mm in the gCAP and VE groups, respectively.

After polyp measurement using the forceps, the mean polyp sizes were 4.0 ± 1.7 mm and 4.5 ± 2.2 mm in the gCAP and VE groups, respectively. The smallest polyp was 1 mm, and the largest was 11 mm.

DP was significantly lower in the gCAP group than in the VE group (gCAP vs. VE, 13.0 ± 22.0% vs. 18.5 ± 19.7%, *p* = 0.029) (Figure 4). Moreover, with 20% and 33% limitation, ER was significantly lower in the gCAP group (gCAP vs. VE with 20% limitation: 24.3% vs. 42.1%, *p* = 0.002; gCAP vs. VE with 33% limitation, 14.3% vs. 27.1%, *p* = 0.008) (Figure 5). With 20% limitation of error, polyps were underestimated with VE than with gCAP (*p* = 0.004) (Table 2).

### 3.2. Subgroup Analysis

Subgroup analysis according to a polyp size of 5 mm was performed in addition. Polyps ≥ 5 mm and ≤5 mm were 99 and 181, respectively. In the group of ≤5 mm polyps, no difference was observed between gCAP and VE. However, gCAP was more accurate for polyps of ≥5 mm (gCAP vs. VE with 20% limitation: 81.8% vs. 49.1%, *p* = 0.001; gCAP vs. VE with 33% limitation: 93.2% vs. 72.7%, *p* = 0.009) (Table 3).

### 3.3. Time Taken for Polyp Size Measurement

The average time required for polyp size measurement was longer in the gCAP group than in the VE group by approximately 4 s (gCAP vs. VE: 8.2 ± 4.8 vs. 4.2 ± 1.5 s, *p* < 0.001). However, the forceps method required 28 s more in both groups (28.1 ± 10.7 s in gCAP and 29.5 ± 13.3 s in VE) (Figure 6).

## 4. Discussion

Even small polyps of ≤5 mm are recommended to be removed, as there could be a 1% chance of an advanced histology [18]. The European Society of Gastrointestinal Endoscopy guidelines recommend different methods of polyp removal according to the polyp size: cold forceps polypectomy for 1–3 mm polyps and cold snare polypectomy for 4–7 mm polyps and 6–9 mm sessile polyps [4]. However, the follow-up interval is based on the polyp size, which can be as large as 1 cm. However, several studies have demonstrated that almost 50% of polyps regarded as advanced based on endoscopic estimates of 1 cm were smaller on pathologic measurements [5,19]. Therefore, it is essential to precisely measure small and diminutive polyps, as well as large polyps (over 1 cm), because this measurement could assist the choice of a polypectomy technique and aid the decision of a follow-up interval.

However, it is difficult to accurately measure polyp size. Although pathologic size measurement has been the “gold standard”, it is not always precise due to incomplete polyp resection; piecemeal resection of polyps, especially using forceps; postfixation deformity of extracted polyps; and interobserver variation of in situ measurements [20,21]. 

Moreover, the decision to remove polyps detected during colonoscopy and the choice of the removal technique are made intraoperatively. Therefore, on-site polyp size measurement is essential.

VE has been widely used in the measurement of polyps, because it takes a shorter time and does not require additional devices. However, VE is known to be inaccurate. Devices such as forceps and a ruler can provide an accurate polyp measurement but may require an additional 30 s or more and may incur additional charges. Therefore, we applied the use of gCAP as a simple and cheap method for polyp size measurement. The gCAP measurement requires only a colonoscope cap and a transparent paper with a grid of 1 mm intervals. 

In this study, the gCAP measurements took 4 s longer than those by VE. However, forceps measurements, which is widely accepted as the standard, required 28 s more than gCAP measurement. Therefore, gCAP measurement is more accurate than VE and is more convenient than forceps measurement.

Many methods such as ruler snares, ruler forceps, polyp sizing posters, laser grids, and calibrated hood indicators have been developed to increase the accuracy of on-site polyp size measurement [9,10,11,12,22,23]. Using devices such as ruler snares or ruler forceps through the working channel could involve additional procedure time than VE [9,10,11]. Additionally, disturbance of the visual field can occur when a calibrated hood indicator is used [12]. We believe that automatic measurement using a laser grid would be the ideal method. However, this method has been evaluated only in an animal experiment, and the laser device may be expensive [22]. Therefore, compared with the other devices, the gCAP method is less expensive and time-consuming and the cap does not disturb the observation field. 

Some studies have demonstrated that repeated polyp size measurement using tools is helpful in training for accurate visual measurements [24,25]. Therefore, we believe that the gCAP method may be useful in the training of beginner endoscopists for improved VE ability.

This study had several limitations. First, polyps larger than 11 mm could not be measured because of the cap size. Second, this was a single-center trial; therefore, the results of a multi-center research and validation are required. Finally, differences between expert and trainee endoscopists were not evaluated. We plan to include the gCAP method in our institutional training course to develop the capacity of our trainees.

In conclusion, on-site measurement of small polyps is important during colonoscopy, and the gCAP method is considered a more accessible and cheaper method for more accurate measurement of the polyp size.

Trial registration: KCT0000754 (Clinical research information service (CRIS) number).

## Figures and Tables

**Figure 1 jcm-10-02365-f001:**
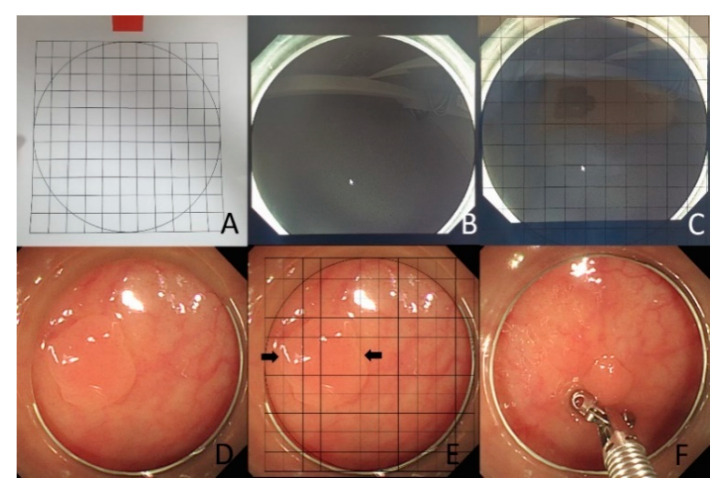
(**A**) The grid was drawn on a transparent vinyl paper with 1-mm intervals, (**B**,**C**). the grid paper was fixed at the inner circle of the colonoscopic cap on monitor, (**D**,**E**). When detected a polyp, the endoscopist attached the cap and measured the polyp size by counting on the external grid, (**F**). In this study, we additionally measured the polyp size with forceps.

**Figure 2 jcm-10-02365-f002:**
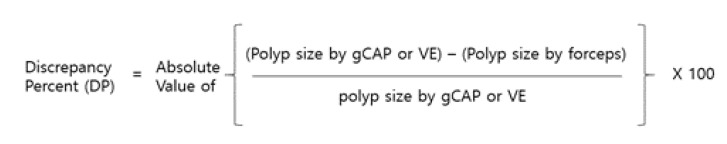
Formula for calculating discrepancy percent (DP).

**Figure 3 jcm-10-02365-f003:**
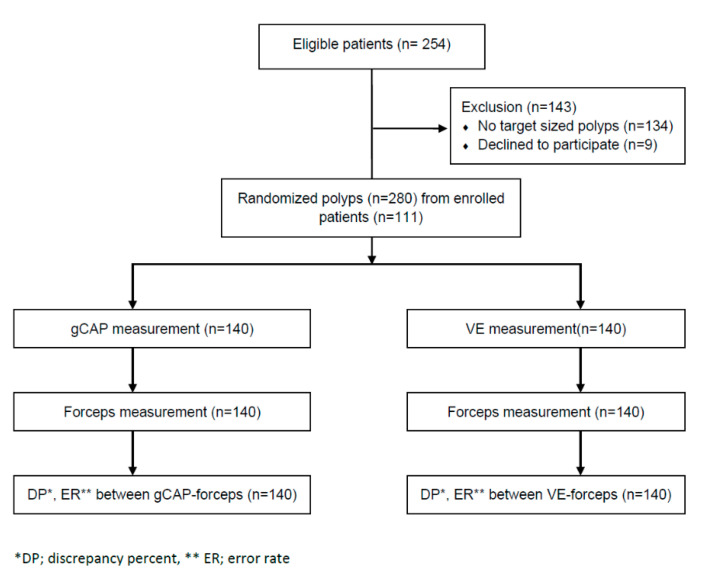
Study process flowsheet.

**Figure 4 jcm-10-02365-f004:**
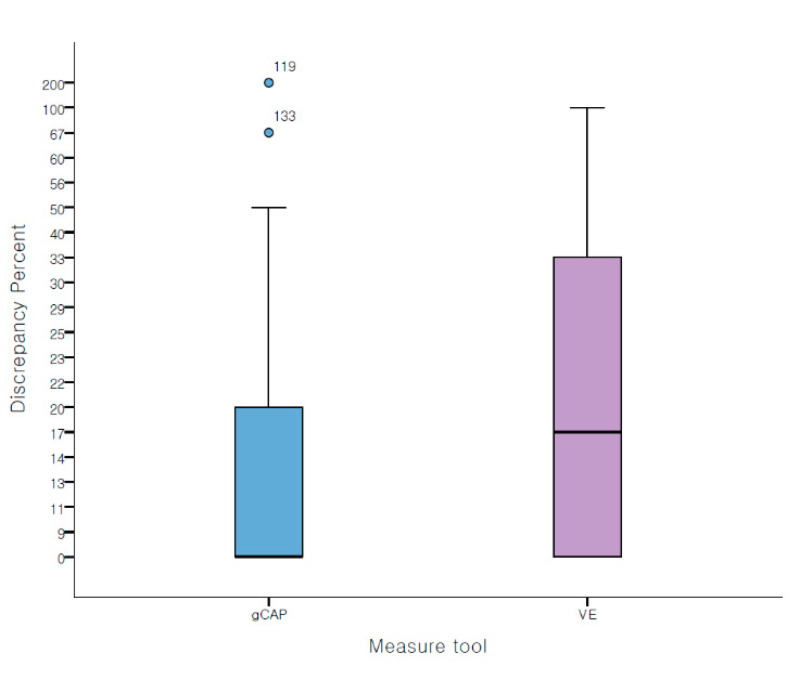
Discrepancy percentages in both groups.

**Figure 5 jcm-10-02365-f005:**
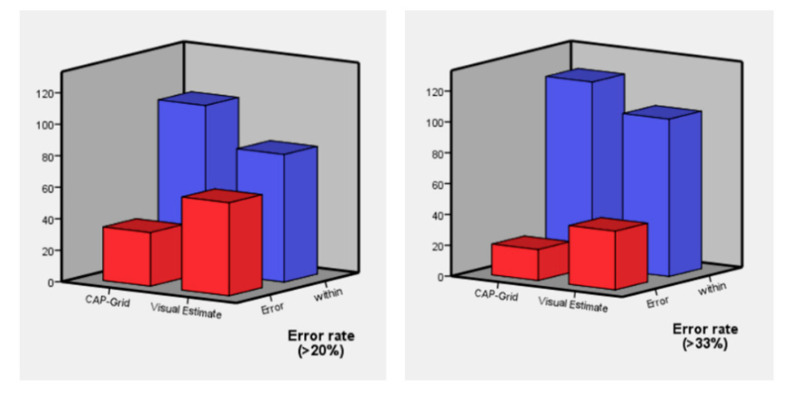
Error rates with limitations of 20% and 33%.

**Figure 6 jcm-10-02365-f006:**
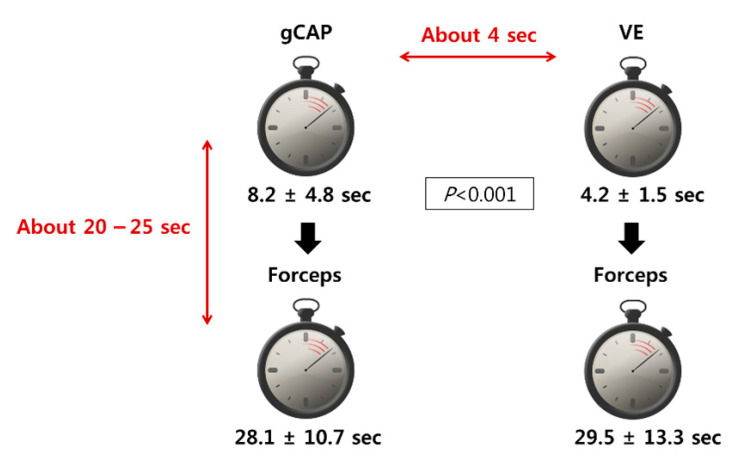
Polyp measurement time using a grid cap, visual estimation, and forceps.

**Table 1 jcm-10-02365-t001:** Demographic and procedural data in both groups.

	gCAP (n = 56)	VE (n = 55)	*p* Value
Age	63.7 ± 8.5	63.8 ± 10.9	0.971
Male sex	71.4% (40/56)	50.9% (28/55)	0.027
Abdominal surgical history	12.5% (7/56)	16.4% (9/55)	0.562
Stomach	5.5% (3/56)	0% (0/55)	
Colon	3.5% (2/56)	7.3% (4/55)	
Gynecology	3.5% (2/56)	9.1% (5/55)	
Adequate bowel preparation	91.1% (51/56)	90.0% (50/55)	0.976
Sedation	66.1% (37/56)	92.7% (51/55)	0.001
Cecal intubation time (sec)	363 ± 235	382 ± 215	0.664
Withdrawal time (sec)	652 ± 292	733 ± 403	0.227
Total number of polyps	2.9 ± 3.1	3.4 ± 2.6	0.389

**Table 2 jcm-10-02365-t002:** Polyp size discrepancy in measurements using a colonic cap with an external grid and by visual estimation.

	gCAP (n = 140)	VE (n = 140)	*p* Value
Mean size using VE or gCAP	4.0 ± 1.7 mm	4.2 ± 1.7 mm	0.368
Mean size by forceps	4.0 ± 1.7 mm	4.5 ± 2.2 mm	0.329
DP * (%)	13.0 ± 22.0	18.5 ± 19.7	0.029
ER with 20% criteria (%, n)	24.3% (34/140)	42.1% (59/140)	0.002
ER with 33% criteria (%, n)	14.3% (20/140)	27.1% (38/140)	0.008
Overcall rate (%, n)	11.4% (16/140)	15.7% (22/140)	0.295
Undercall rate (%, n)	12.9% (18/140)	26.4% (37/140)	0.004

* DP: discrepancy percent, ER: error rate, gCAP: colonic cap with an external grid, VE: visual estimation.

**Table 3 jcm-10-02365-t003:** Subgroup analysis according to polyp size.

	Subgroup Analysis		
1–4 mm	gCAP (n = 96)	VE (n = 85)	*p* value
DP (%)	12.9 ± 15.5	14.2 ± 16.1	0.554
ER with 20% criteria (%, n)	27.1% (26/96)	37.6% (32/85)	0.129
ER with 33% criteria (%, n)	17.7% (17/96)	27.1% (62/85)	0.130
≥5 mm	gCAP (n = 44)	VE (n = 55)	*p* value
DP (%)	13.4 ± 32.1	25.0 ± 22.8	0.037
ER with 20% criteria (%, n)	18.2% (8/44)	49.1% (27/55)	0.001
ER with 33% criteria (%, n)	6.8% (3/44)	27.3% (15/55)	0.009 *

* DP: discrepancy percent, ER: error rate, gCAP: colonic cap with an external grid, VE: visual estimation.

## Data Availability

Not applicable.

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
