# Peer review of "Usefulness of a Colonoscopy Cap with an External Grid for the Measurement of Small-Sized Colorectal Polyps: A Prospective Randomized Trial"

_jcm, 2021, doi:10.3390/jcm10112365_

Round 1

Reviewer 1 Report

Thank you for this interesting endoscopy/colonoscopy study. I have to admit that the manuscript is well-written and aside from some comments with the tenses (e.g., the diameter for the cap was 11mm, it has to be corrected to is, as the diameter of the study object remains the same before during and after your study) I didn't find any other problems in its grammar presentation.

Having said that, figure 1 is only abstract, and I would expect that you will present the whole steps of the study not only the drawing up of the grid circles but where/how was applied on the cap, monitor etc so one can follow your very visual study without even consulting the text. That will be a success that will enhance the visibility and any potential citatability of your paper.

I recommend that you apply this 'rule' for the rest of your images.

Author Response

  1. This is an interesting study showing that the gCAP method is significantly more accurate for estimation colon polyp size compared to visual evaluation (VE). The study design is adequate. However, it remains questionable whether the findings have significant clinical impact because the mean polyp size was not significantly different between gCAP and VE. In addition, the additional costs of gCAP in relation to its clinical benefit appears to be critical and should be discussed.

Answer) I fully agree with you. The primary outcome was accurate polyp size measurement. Firstly, we measured mean difference of polyp size between gCAP/VE and forceps. The mean difference included overcall and undercall. So, we adopted the discrepancy percent (DP) which showed significant difference.

These results means that the gCAP could measure more precisely than VE on site.

However, I don't agree for routine use of colonic cap to measure size of every polyps with more 10 to 20 US dollar charge. Some studies have demonstrated that repeated polyp size measurement using tools is helpful in training for accurate visual measurements. Therefore, I believe that the gCAP method may be useful in the training of beginner endoscopists for improved VE ability. So, I plan to include the gCAP method in our institutional training course to develop the capacity of our trainees. I described this point on discussion session.

Thank you very much.

Reviewer 2 Report

This is an interesting study showing that the gCAP method is significantly more accurate for estimation colon polyp size compared to visual evaluation (VE). The study design is adequate. However, it remains questionable whether the findings have significant clinical impact because the mean polyp size was not significantly different between gCAP and VE. In addition, the additional costs of gCAP in relation to its clinical benefit appears to be critical and should be discussed.

Author Response

(The authors gave the same response as above.)

Round 2

Reviewer 2 Report

After revision the manuscript is now appropriate for publication in JCM.